# Secured Key Distribution by Concatenating Optical Communications and Inter-Device Hand-Held Video Transmission

**Menachem Domb \* and Guy Leshem**

Computer Science Department, Ashkelon Academic College, Ashkelon 52653, Israel; leshemg@cs.bgu.ac.il
\* Correspondence: dombmnc@edu.aac.ac.il

**Abstract:** Key distribution is a growing concern for symmetric cryptography. Most of the current key-distribution mechanisms assume the use of the Internet and WAN networks, which are exposed to security hazards. To overcome them, the use of comprehensive and robust cryptographic mechanisms such as Diffie–Hellman (DH) and RSA (Rivest–Shamir–Adleman) algorithms are proposed. These solutions are limited. DH and RSA have been under threat since the introduction of quantum computing. Hence, new ideas are required. This paper introduces a new security approach for a safe key transmission using undetected high-speed optical camera communication (OCC), which is based on the rolling shutter effect of modern smart cameras. The key transfer is done by blinking light-emitting diode (LED) lights in a specific sequence and frequency, following an encoding pattern. The receiver decodes the received blinks to a bit string using a corresponding image processing application. This optical media ensures secure transfer without the ability to quote it. We demonstrate in an experiment the feasibility of using basic wireless optical communication for key transmission, eliminating the need for a permanent, long, and costly setup. It is mobile, available everywhere anytime, and requires only simple connections to operate. The results show that this method is feasible, robust, efficient, and implementable.

**Keywords:** encryption key; secured key distribution; optical communications; symmetric and asymmetric encryption

## 1. Introduction

Extra efforts are required for protecting the processing of sensitive data. These security efforts should follow the entire life cycle of a given data element, such as records, files, and databases. Cryptographic techniques are the most popular tools used to protect data during its life cycle, starting from its generation through to its transmission via public communication networks, being saved in the database, and finally throughout its maintenance period. The most common implemented cryptographic tool is symmetric cryptography, where the same secret key is used to encrypt and decrypt the data. The main risk of this approach is the possibility of discovering the secret key while it is exchanged between the two communicating parties. To avoid this, we are required to frequently replace the encryption key. Since the key exchange process is a very sensitive and critical stage, we must employ a higher security level at the key exchange phase. There are many known solutions, such as the specific key exchange protocol proposed by Diffie–Hellman or encrypting the key using asymmetric cystography such as Rivest–Shamir–Adleman (RSA) algorithms. However, with the growing accessibility of quantum computing, these solutions are losing pace over time.

Another approach is by establishing a reliable third party who generates certificates and encryption keys and simultaneously distributes them to the two parties who intend to exchange data. These

parties use alternate distribution channels that are different from the channels the two parties use to transfer the data. However, due to the growing globalization and increasing distance among users and systems and the introduction of cloud computing, this approach is very complicated to manage and thus has become irrelevant over time. Apparently, we need to search for more alternatives for key exchange. A robust, highly secured, simple, and easy-to-implement solution that overcomes these expanding constraints is preferable.

We propose a complete and secured cycle for the secured distribution of the generated key. In the first stage, an encryption key is generated in a secured processing environment, and in the second stage, the key transmission uses an optical communication platform.

The rest of this paper consists of the following sections. Section 2 outlines optical communications relevant to our proposed solution. In Section 3, we describe the main components comprising our solution. In Section 4, we present the overall proposed solution, and in Section 5, we detail the developed application and the experiment we conducted and its results. We conclude in Section 6, with a discussion and directions for future work.

## 2. Related Work

Optical communication is a simple, low cost, and secured signal transmission. One of the technics is based on under-sampled differential phase shift on–off keying that can encode binary data. Arai et al. [1] define a new framing approach for high-speed optical signals transmission for road-to-vehicle communication. This technique can be used as is for our solution implementation especially due to its high-speed communication. Luo et al. [2] propose a basic approach to increase the transmission rate using direct unwired optical communications by means of dual light-emitting diodes (LEDs), resulting in triple the data rate transmission. Roberts [3] proposes a new encoding/decoding approach by synchronizing the camera subsampling with the camera frame rate. This improvement is effective, as the frame rate is constantly improving in each new version of digitized cameras, especially in cameras embedded in mobile devices. Leu et al. [4] introduce a new modulation scheme where the phase difference between two consecutive samples represent one-bit data. This new feature enhances the communication speed and may be used as an alternate or backup modulation in case the alternate modulation incurs a problem. A comprehensive analysis and implementation considerations of the optical camera communication (OCC) technology using smartphone cameras is detailed by Shahjalal et al. [5]. Their research demonstrates an OCC system using a low frame rate smartphone camera to analyze the requirements and critical implementation challenges. A variety of solutions are described to demonstrate improvements communication capacity, distance, and stability.

## 3. Technologies Used in the Proposed Solution

The optical communication technique called optical camera communications (OCC) is described in [5,6]. OCC allows the use of huge unregulated bandwidths in the optical domain spectrally located between microwave and X-ray wavelengths, as shown in Figure 1. In such systems, an image sensor and a camera are used to demodulate the transmitted signal, which has been modulated according to on–off keying (OOK). Current available devices are smart devices equipped with LED flash and cameras. These provide a pragmatic form of an Optical Wireless Communication (OWC) where LED projectors deliver the Visible Light spectrum (VLC) component and a camera as the receiving module, creating a transceiver pair.

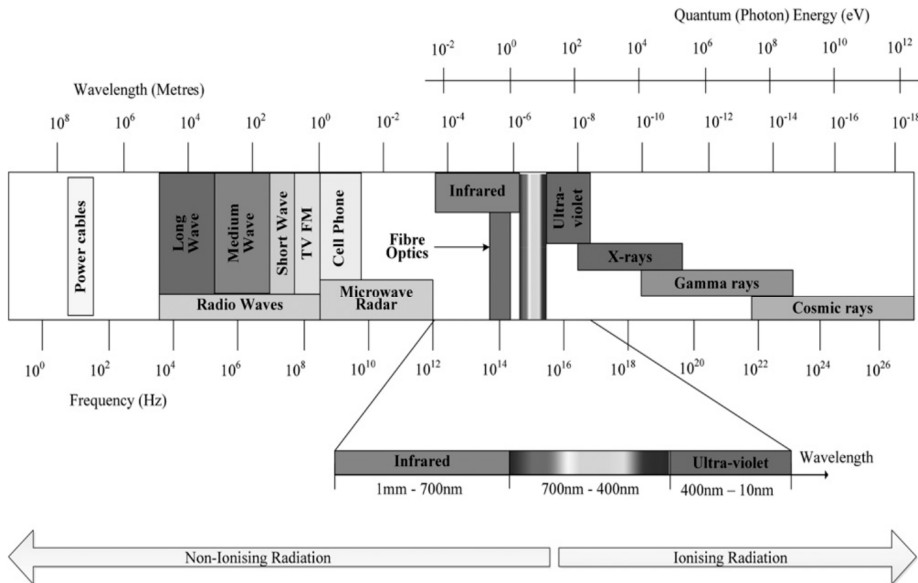

**Figure 1.** Electromagnetic spectrum range.

The OCC system uses commercial LED lighting sources that include LED-based infrastructure lighting, LED flashes, LED tags, displays, laser diodes, image patterns, and some current generation projectors. The major driving forces of OCC deployment are the widespread availability of visible light (VL) LEDs and the possibility of utilizing the camera in the smart devices to decode LED modulated data. Therefore, these LED infrastructures can be used for data transmissions using on–off keying (OOK).

A typical OCC system is shown in Figure 2, where a camera is used as a receiver that consists of an imaging lens, an image sensor, and a readout circuit.

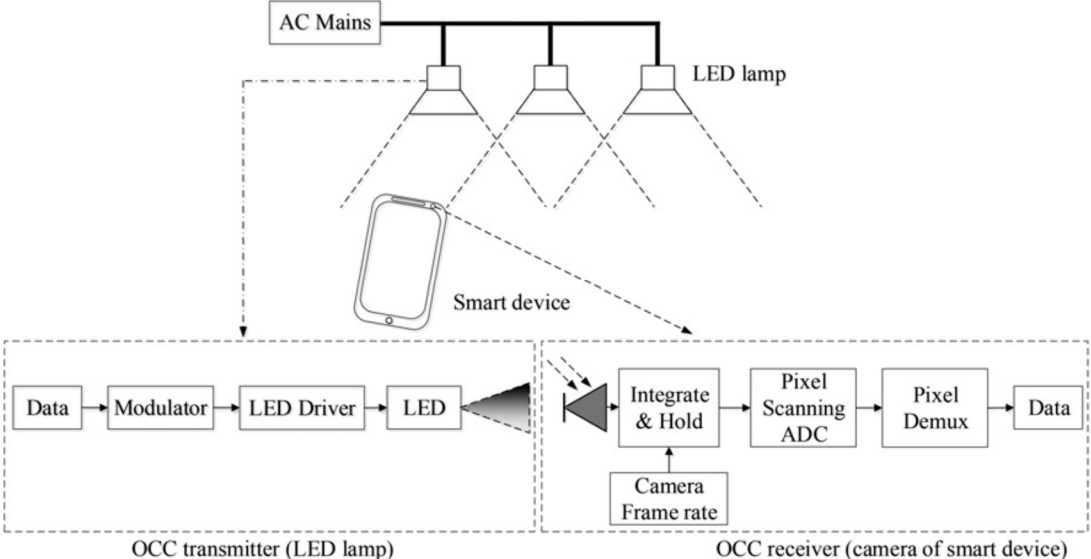

**Figure 2.** A schematic view of the optical camera communication (OCC) system.

Optical communications comprise an LED, Infra-Red or Lazier projector, and a high-speed camera embedded in a mobile phone. The projector projects a beam of light toward the direction of the camera. The camera has an embedded CMOS image sensor that captures the projected beam. The beam on/off projection duration and frequency is according to an encoded pattern coordinated with the receiving camera. The receiving camera records a video of the projection session and saves it in its internal

storage. The recorded projection can be decoded into bits, where for an "on" beam, the corresponding decoding bit is set to "1", and otherwise, it is decoded as "0". The video in the camera can be further transmitted to the target receiver through a public network connection. The beam may be visible (normal lighting) or invisible (Infra-Red and Lazier). When the CMOS image sensor is operated, images are captured. These images are the source for extracting data by decoding it. Figure 3 depicts the three phases of the received signal processing. The left image is the original recorded beam impact, the one to the right is the original image after it was crystalized, and the third image to the right depicts the final stage of the process. The third image is the input for decoding the beam stream into a bit string.

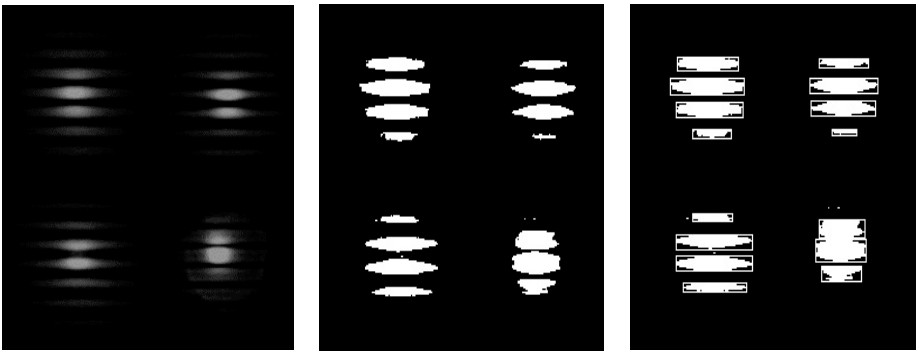

**Figure 3.** Three phases of fringe signal processing (Dibbiny and Kays, 2017).

Figure 4 depicts the encoding process, starting with processing the image and translating it into a sequence of a signal chart (the top chart). The bottom chart depicts the final bit sequence.

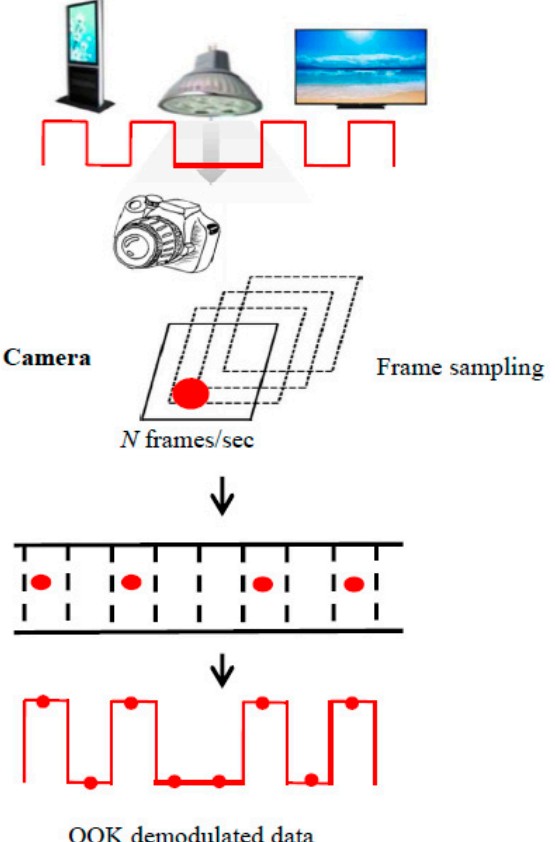

**Figure 4.** Optical camera communication architecture.

### 4. Overview of the Proposed Solution

The goal of this work is to enhance the security of key generation and key distribution. We propose key generation in an ultimate secured processing environment provided by a secured development environment with an alternate and highly secured communication protocol by means of an optical communications technique. With this method, the exchange of keys is done by an optical means of communication, where the LED transmits, and the camera captures and records it. The idea is to modulate the information in a way that is complicated to be detected, and the information can be decoded only by processing the appropriate signals of the video camera.

Figure 5 illustrates the basic idea of the optical-based communication approach. The source computer generates the encrypted key. The key is translated to optical signals, where "1" means the LED is "on" and "0" means it is off. The LED projector keeps projecting optical signals to the camera, which is embedded in a mobile device. The camera captures the transmitted optical signals and records it as a video movie. For authentication and accuracy, the video movie is signed by a standard electronic signature. For an enhanced security level, the signed video is encrypted by an asymmetric cryptography used just for key wrapping encryption and transmitted via VPN to the target mobile device. At the target, the transmitted key is unwrapped and projected to the target camera, which decodes it and sends the key bit string to the target computer, and thus the key reaches the target computer. Then, the target computer decrypts the encrypted messages. To enhance the security, we can consider the possibility of moving the mobile device itself toward the target computer and skipping the need to transmit the video between the two mobile devices, which may be considered less secure.

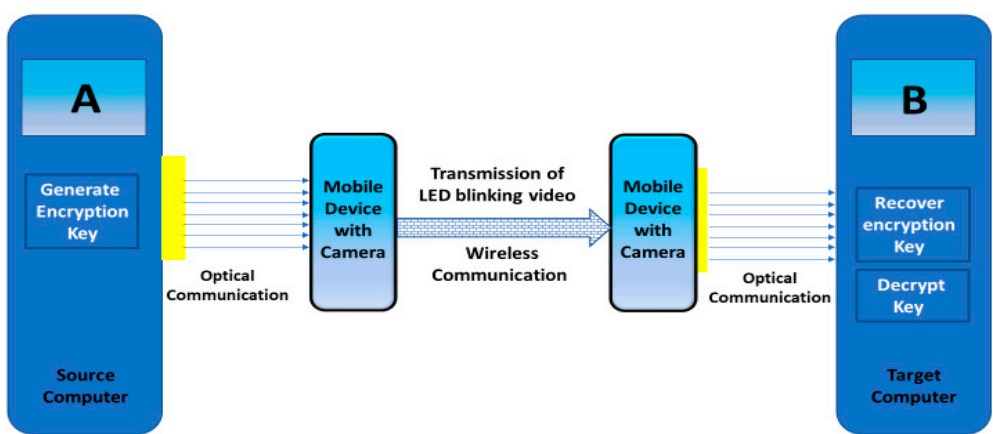

**Figure 5.** High level secured key transmission system.

Figure 6 illustrates the entire flow. The source computer, in Step 1, generates the encryption key. In Step 2, it converts the encrypted key into an optical code, which then is projected by its attached LED projector to the mobile camera that is embedded in the sender's mobile device as a sequence of LED images. In Step 3, the source camera accepts the LED images, and the mobile device transmits the accepted video to the target mobile device. In Step 4, the receiver camera projects the received LED images to the optical receiver at the target computer. Then, the images are copied to the secured environment of the same computer, which, in Step 5, translates the images into a bit string, representing the encryption key. Subsequently in Step 6, the decryption process is applied to obtain the original key. Then, the key is forwarded to the regular environment for further processing.

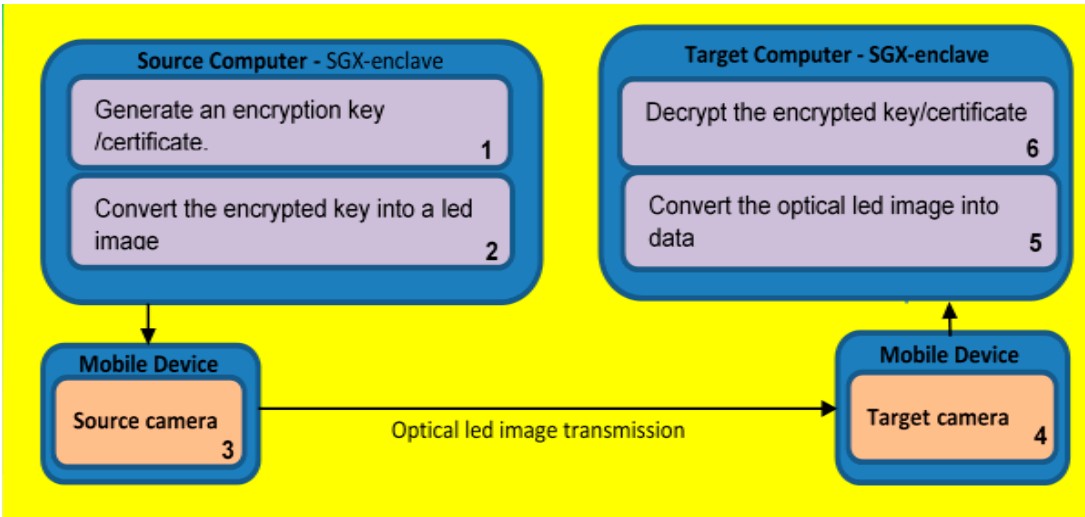

**Figure 6.** Encryption key generation and optical-based transmission flow.

Figure 7 depicts the transmission protocol and messaging flow between the mobile device and the computer.

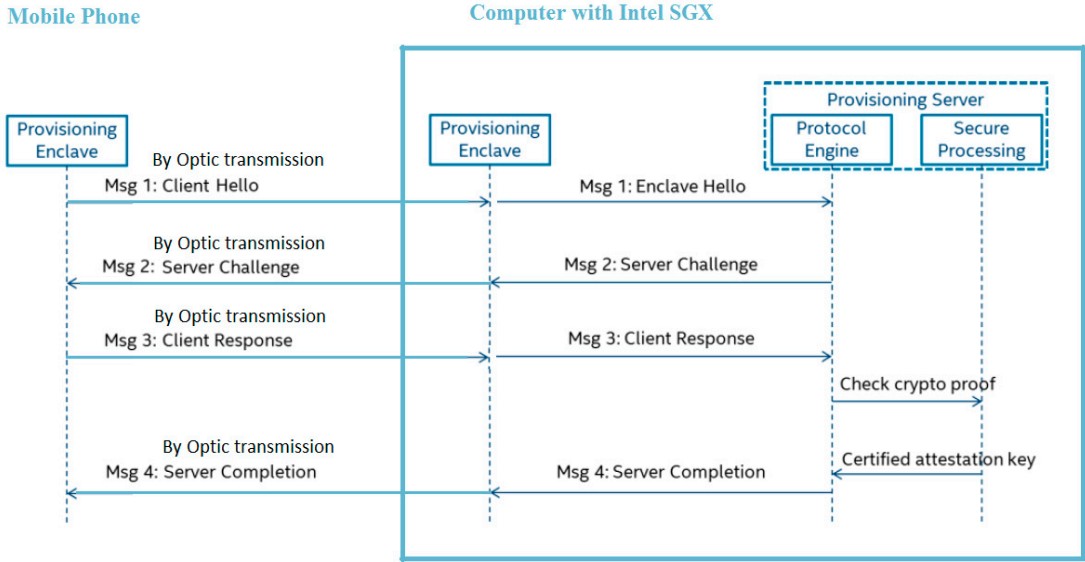

**Figure 7.** Optical-based provisioning message flow protocol.

The proposed solution of transmitting video signals instead of transmitting the original bare bit string of the key adds computational processing complexity that is equal to the standard image processing complexity. The volume of transmitting video/images is larger in magnitude from a common encrypted bit stream. However, considering the limited key length and the advances in transmission capacity and speed, it is still justified is comparison to the security level it adds.

Looking at the proposed transmission route, most of it uses standard communication lines and is considered as protected from a potential adversary. However, the optical transition stage is still at risk, as optical signals can be captured from remote using appropriate technologies. To avoid it, we request the optical-signal transmission to be a short distance from the projector and within a non-transparent cover.

## 5. Experiment and Results

In this section, we describe the hardware and the associated software code involved in enabling the secured key generation and transmission. We begin with the key generation phase and continue with the transformation to optical signals and the rest of the implementation. Testing the correctness and functionality of the proposed solution was successfully executed, and it passed all the functional tests. For non-functional testing, we revealed a reduced pace of the transmission process. This is mainly due to the transmission via different medias, bit-wise and optical-wise, and the immense difference in the transmitted volume. However, since the key size is limited, we did not notice an impact that would cause us to classify this solution as impractical.

### 5.1. Generating the Encryption Key

For simplicity, we generate a new key by activating a function that produces a random number with the desired length, and then transmit it, securely, by the optical means.

For illustration purposes, we present the code in Figure 8:

```cpp
unsigned int generate_key(size_t key_size_in_bytes)
{
    unsigned char  * keys ;
    sgx_status_t status;

    keys = new (nothrow) unsigned char [key_size_in_bytes];
    if (keys == nullptr)
        return 0;

    status = sgx_read_rand(keys, key_size_in_bytes);

    if (status != SGX_SUCCESS) return 0;

    return 1;

}
```

**Figure 8.** The code for generating the Encryption Key.

### 5.2. Transforming the Encryption Code into Blinking LED

We used Linkit ONE hardware. Figure 9 depicts the blinking LED controlled by an Arduino code and connected to computers via a plugged-in USB.

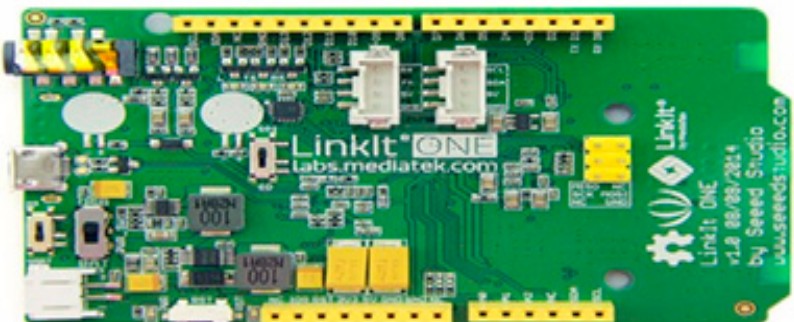

**Figure 9.** Linkit ONE hardware.

The test bed assumes proper security measures applied for data transmission from the computer memory to the USB. To transfer the generated encryption key, we convert it into a bit sequence. The encoding of the key into a blinking sequence can be done using a variety of formulas, which may be complicated to decrypt. However, for simplicity, we used a standard encoding/decoding algorithm, where for a "1" bit the LED blinks for a very short specific time, and otherwise it is kept dark. The code in Figure 10 outlines the blinking operation of the LED lamp.

```
void serialEvent()
{
    while(Serial.available())
   {
       int ch = Serial.read();
       Serial.write(ch);
       if (ch != -1) {
               switch(ch) {
                  case '0':          // turn off D13 when receiving "0"
                            digitalWrite(13, 0);
                            break;
                     case '1':          // turn on D13 when receiving "1"
                            digitalWrite(13, 1);
                            break;
```

**Figure 10.** The blinking operation of the LED lamp.

*5.3. Transferring the Encryption Key to the USB*

The encryption key is transmitted to the USB device. The ***Serial.exe*** program receives the key and converts it into a binary sequence according to an ASCII code. To signal the start of a new key transmission, an 0xA5 bit string is sent. The program code in Figure 11 uses a serial object that contains features and functions allowing communication via USB ports.

```
class Serial
{
   private:
      HANDLE hSerial; //Serial comm handler
      //Connection status
      bool connected;
      COMSTAT status; //Get various information about the connection
      DWORD errors; //Keep track of last error
   public:
      //Initialize Serial communication with the given COM port
      Serial(const char *portName);
      //Close the connection
      ~Serial();
         //Read data in a buffer, if nbChar is greater than the maximum number of bytes
         available, it will return only the bytes available. The function returns -1 when
         nothing could be read, the number of bytes read.
      int ReadData(char *buffer, unsigned int nbChar);
      //Writes data from a buffer through the Serial connection return true on
            success.
```

**Figure 11.** The code for transferring the Encryption Key to the USB.

Figure 12 depicts the transmission of a 32-bit size "abcd" key.

.1.0.1.0.0.1.0. .1.0.0.0.0.1.1.0 .0.0.1.1.0 0 .0.1. .0.0.1.1.0 0 .1.1. .0.0.1.1.0 1 0.0.

**Figure 12.** Example of the key transmission.

*5.4. Transmitting the Blinking LED Video to A Remote Receiver*

The transmission of the blinking LED video to a remote receiver is done via the internet. To ensure a safe and secured transmission, we used a light version of the Chaotic key stream cipher executed during real-time video communication proposed by Gilles Millérioux et al. [7]. It embeds cryptography in real-time video streaming using a symmetric cryptographic algorithm (DES). With the same key on both sides, the video is correctly displayed on the receiver's side with a reasonable bit rate. The received video is properly displayed on the receiver's display.

*5.5. Transforming the Blinking LED into Data*

We developed an image processing application, which accepts the sequence of the blinking LED, processes it to produce a bit sequence, and converts it into an ASCII code. A lit LED is processed by the openCV image processing in the following manner: the image is blurred using the medianBlur. Each non-white pixel turns black (0), while white remains (255). At the end, all the pixels' values are summed. If the sum is 0, the LED remains "off", and the output is a "0" bit. Otherwise, the output is a "1" bit. The code in Figure 13 outlines the translation of a blinking LED into a bit string. An explanation of the code appears in the program lines.

```
public Mat process(Mat rgbaImage) {
    // Convert the image to Gray
    Mat grayImage = new Mat();
    Imgproc.cvtColor(rgbaImage, grayImage, Imgproc.COLOR_BGR2GRAY);
    //blur image to ignore noise
    Mat blurImage = new Mat();
    Imgproc.medianBlur(grayImage, blurImage, 11);

    //detect white space in image - LED white balance
    Mat tImage = new Mat();
```

**Figure 13.** *Cont.*

```java
Imgproc.threshold(blurImage, tImage, 245, 255, 0);

Scalar imageSumElems = Core.sumElems(tImage);

isLedOn = false;

for (int i = 0; i < imageSumElems.val.length; i++)
    if (imageSumElems.val[i] > 0)
        isLedOn = true;
return rgbaImage;
}
```

**Figure 13.** Translation of a blinking LED into a bit string.

### 5.6. Compliance with NIST Standards and Guidelines [SP 800-57 Pt. 1 Rev. 4]

Key management provides the foundation for the generation, storage, use, distribution, and destruction of keys [2]. For secured key generation and storage, we used an Intel SGX secured environment. Then, the key is moved to the USB-connected blinking device for encoding the key to a blinking LED stream. We used a handy blinking LED device attached to the sender computer via a USB connector. According to the device specifications, the blinking speed can be up to 1000 blinks per second. However, the collecting smartphone camera we used is limited to 30 blinks per second. Therefore, we have tuned the blinking device to 30 images per second, which seems to be enough for the purpose of our experiment to prove the feasibility of using basic wireless optical communications for key transmission, eliminating the need for a permanent, long, and costly setup. It is mobile and available everywhere anytime, requiring simple connections to operate. It is similar to the described OC system by Xuhua in [8]. The experiment of a key distribution prototype complies with the NIST guidelines outlined in [9] Sections 8.1.5.2.2.1 and 2.

According to these guidelines, during the distribution, the key must be protected throughout the entire distribution via a communication channel. Only approved key-wrapping or public key-transport schemes shall be used. For symmetric key-wrapping schemes, the distributed key is not disclosed or modified. For an asymmetric key-transport schemes, the private key and the distributed key are not disclosed or modified, and the parity of private–public keys is maintained. The keys are protected in accordance with the guidelines for protection requirements for cryptographic information. It should provide the following assurances: each entity in the distribution process knows the identifier associated with the other entity, the keys are correctly associated with the entities involved in the key-distribution process, and the keys have been received correctly. Our experiment complies with these guidelines, as it is practically embedded already as a standard in current implementations. In our experiment, we comply with these guidelines by encrypting the encoded key (blinking LED) by an asymmetric cryptography used just for key wrapping. For extra protection, we use VPN for encoded key distribution.

### 5.7. Testing the Experiment Prototype

To ensure the feasibility and integrity of the experiment prototype, we set the criteria for passing/failing the functional and non-functional testing. We conducted a series of test activities. We began with unit testing each component participating in the chain of activities, including encryption key generation, transformation to blinking LED video, securely sending the video to the receiver,

and transformation of the optical signals back to the key bit string. The testing included test cases for each component, such as checking the correctness of the output after each transformation stage. Then, we prepared system test cases to ensure the correctness and smooth key transmission from the sender's processor to the acceptance of the key by the receiver's processor. The system met the predefined criteria.

*5.8. Execution and Results*

To prove our proposed solution, we ran the entire process cycle. The encryption key was generated in a secured environment; then, it was transmitted a bit string to the USB blinking device. The mobile phone camera recorded the video of the blinking sequence. The mobile phone signed, encrypted, and transmitted the blinking LED video. The mobile receiver device in the target location accepted the blinking video and transformed it into a bit string. Then, the bit string was translated into the ASCII code to retrieve the original encryption key. This key was used to decrypt the corresponding transmitted messages.

We performed an experiment of a key transfer (e.g., "a b c d") between a host with an optical USB device and a smartphone with a camera, where each character had 8 bits.

Figure 14 depicts the key transmission example ["abcd"] used in the experiment. The four screen images were captured during the live key transmission stages. In image a, the stream of the 0xA5 bit string was accepted by the mobile device connected to the sending computer. Image b captured the sender computer screen shot during the key transmission to its associated mobile device. Image c is the captured mobile screen while accepting the "abcd" key and image d depicts the complete acceptance of the transmitted key.

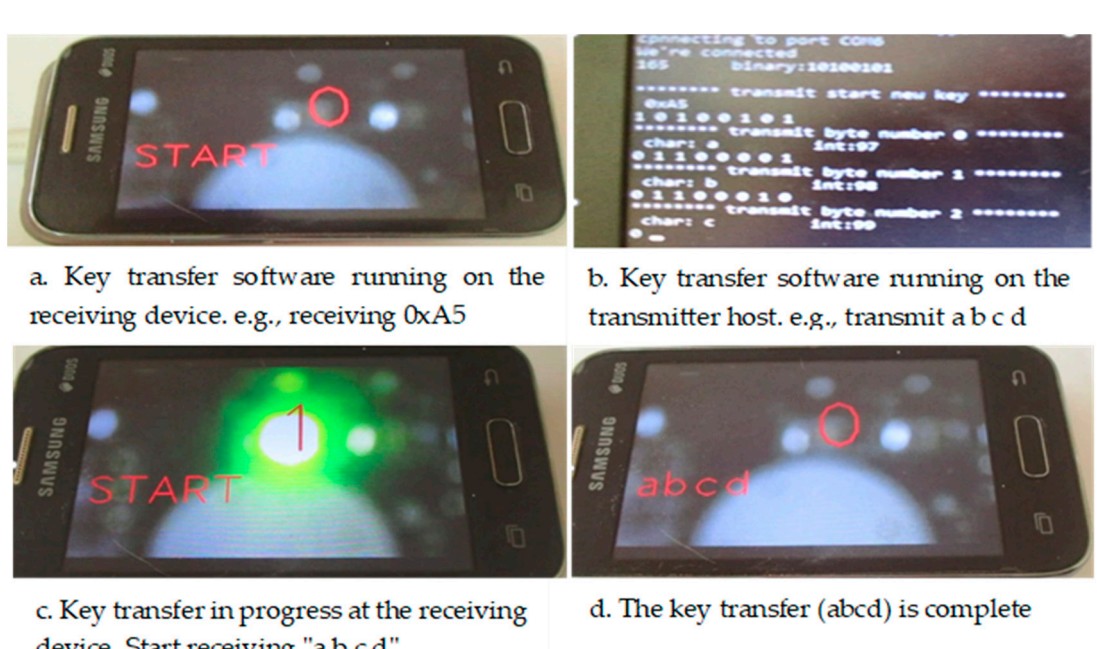

a. Key transfer software running on the receiving device. e.g., receiving 0xA5

b. Key transfer software running on the transmitter host. e.g., transmit a b c d

c. Key transfer in progress at the receiving device. Start receiving "a b c d"

d. The key transfer (abcd) is complete

**Figure 14.** Captured images of the live key transmission stages.

## 6. Conclusions

In this paper, we introduced an encryption-key transmission using optical communication. We detailed an experiment of the entire cycle of key transmission, including the employed hardware components. The results of the experiment prove the applicability of our proposed solution, providing an additional tool for secured key management.

We plan to further explore the possibility of expanding the optical transmission capabilities to ensure the secured transmission of sensitive data utilizing advancements in mobile devices and

their camera capabilities. Furthermore, we intend to explore more innovative solutions for key transfer protocols.

**Author Contributions:** The idea of using optical communications for key transmission came up in a discussion between the two authors. The definition of the concept, modeling, designing, studying the related work and composition of the paper was done by the M.D.; Defining and building the test bed, the programming and executing the testing was done by G.L. All authors have read and agreed to the published version of the manuscript.

**Funding:** This research has been funded by the Ashkelon Academy and its publishing has been funded by MDPI.

**Conflicts of Interest:** The authors declare no conflict of interest.

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
