# Peer review of "Secured Key Distribution by Concatenating Optical Communications and Inter-Device Hand-Held Video Transmission"

_asi, doi:10.3390/asi3010011_

Round 1

Reviewer 1 Report

Algorithms can be presented much more compact and more understandable in mathematical form and not as source code listings.

There is no reason to mention Intel SGX several times because it does not provide any importance for the research result.

The article lacks of very important part of any applied security system: adversary model. What kind of risks did you prevent and which does not?

There is no serious comparison with any alternatives for key distribution.

Also no estimations of secure channel throughput, computational processing complexity, amount of extra information used to pass secure bits were provided.

The most important part of secret key travel is not clear (lines 145-147): whether mobile device with secret key video is moved to the target computer's vicinity or not?

Author Response

Dear reviewer 1: Thanks very much for reviewing our paper draft. Following is our reply to the review comments

Algorithms can be presented much more compact and more understandable in mathematical form and not as source code listings.

Thanks for this comment. Indeed, we have removed some of the software code. However, since we utilize a secured environment, which is not commonly used by programmers, we demonstrate how it is easy to incorporate it in standard programming and so increase the awareness of using such technology for highly sensitive and secured data. The code includes explanations within the code.  

There is no reason to mention Intel SGX several times because it does not provide any importance for the research result.

Thanks. We have concentrated SGX into one sub section 3.1. Although the paper focuses on the key transmission, we still prefer to present the entire process, starting from generating the key in a secured environment up to the actual use of the generated key by a remote processor.    

The article lacks very important part of any applied security system: adversary model. What kind of risks did you prevent, and which does not? There is no serious comparison with any alternatives for key distribution.

Thanks. We have highlighted the secured key transmission problem, which current  technologies lack to provide. We mention in the introduction alternate approaches such as Diffie-Helman, RSA, CA etc. and explain its disadvantages.  

Also, no estimations of secure channel throughput, computational processing complexity, amount of extra information used to pass secure bits were provided.

Thanks, we added to the end of section 4, the extra transmission size, which is the difference between a simple bit and a video presenting of this bit. There is not much complexity difference. However, from the transmitted size the difference is significant. Since the key is limited in length the practical impact is minimal due to the advances in communication-channels capacity and transmission speed.  

The most important part of secret key travel is not clear (lines 145-147): whether mobile device with secret key video is moved to the target computer's vicinity or not?

Thanks. We have cleared this point. We mention two ways of the transmission platform of the key from the sender to the receiver computers.

The first is by sending a video with the original translation of the key to a sequence of blinking signals. The video itself is signed and encrypted.

The second option is a physical transition of the mobile phone itself to the receive location.

Both options are feasible. 

Reviewer 2 Report

The paper is interesting, but present some important problems that must be addressed before acceptance:

First, the Introduction is too short. I think the problem to be addressed must be explained with details, as well as the context and how the proposed technology is envisioned to solve all the described situation.

A similar problem is present in Section 2. Several references are provided but they are not discussed enough. The different approaches should be described in a coherent way, describing the existing problems and the differences with the proposed mechanism. 

Section 3 and Section 4 need more technical details. What kind of certificate may be employed, what kind of key... any type? 

Section 5 is probably the main problem of this paper. This describes an initial implementation but no experiment is provided. What research questions are you addressing? what variables are you measuring? methodology? a comparison with existing solutions should be provided. Standadrd security tests should be employed.

 Finally, Section 6 needs to be extended

Author Response

Dear reviewer 2: Thanks very much for reviewing our paper draft. Following is our reply to the review comments

The paper is interesting, but present some important problems that must be addressed before acceptance:

First, the Introduction is too short. I think the problem to be addressed must be explained with details, as well as the context and how the proposed technology is envisioned to solve all the described situation.

The introduction has been extended with a detailed explanation of the current situation of key exchange technics and the problem definition

A similar problem is present in Section 2. Several references are provided but they are not discussed enough. The different approaches should be described in a coherent way, describing the existing problems and the differences with the proposed mechanism.

Thanks, we added to the Related work section more details discussing existing key transmission approaches, solutions and technologies. 

Section 3 and Section 4 need more technical details. What kind of certificate may be employed, what kind of key... any type?

Thanks. Technical details have been added to the relevant sections. Regarding the keys and certificates, we added a clarification saying that when we mention "key transmission" we refer to an encryption/decryption key used for Symmetric Cryptography.

Section 5 is probably the main problem of this paper. This describes an initial implementation, but no experiment is provided.

An extended explanation has been added including the software code some algorithms and screenshots describing the transmission process.

What research questions are you addressing? What variables are you measuring? methodology? a comparison with existing solutions should be provided.

The search question has been clarified in several places in this paper, starting from the abstract. The comparison to existing solutions is outlined in the related work section.

Standard security tests should be employed.

We have added it.

Finally, Section 6 needs to be extended

Thanks. It was extended.

Reviewer 3 Report

The paper presents the possibility of transferring cryptographic keys using optical communications and inter-device hand-held video transmission. In a scenario presented in the paper, smartphones camera capabilities are used.

The idea of the paper is interesting, but the use of such a medium to transfer such sensitive data is questionable. The possibility of message interception is quite high as the way that the key is transferred is not a secure channel. 

Additionally, the chapter describing the proposed approach is very brief. The experimental results are not validated adequately, there is no discussion of the results.

Generally, also the chapter following chapter 4 which describes the solution are very short and make the impression of work in progress.

Author Response

Dear reviewer 3: Thanks very much for reviewing our paper draft. Following is our reply to the review comments

The paper presents the possibility of transferring cryptographic keys using optical communications and inter-device hand-held video transmission. In a scenario presented

in the paper, smartphones camera capabilities are used.

The idea of the paper is interesting, but the use of such a medium to transfer such sensitive data is questionable. The possibility of message interception is quite high as the way that the key is transferred is not a secure channel.

Thanks. In section 4, we added  authentication and security means by employing Signature and Encryption of the transmitted video and proposed the use of VPN. We agree and assume that for practical implementation there is a need to fine tuning the proposed conceptual solution.

Additionally, the chapter describing the proposed approach is very brief.

Thanks. We added more details explaining the proposed solution

The experimental results are not validated adequately, there is no discussion of the results.

Our target at this paper is the introduction of a solution and approve it is feasible and implementable. We continue to work on it, and we'll add performance and other non- functional measurements.   

Generally, also the chapter following section 4 which describes the solution are very short and make the impression of work in progress.

Since the solution required the integration of hardware and software components, it was much more complicated and time consuming. Therefore, we focused on building the right building blocks with the ability to execute the full cycle of generating the key, transmitting the key and decrypt with it the ciphered message. 

The section has been revised accordingly.

Round 2

Reviewer 1 Report

1. Section 1

There are many known solutions, such as a specific key exchange protocol proposed by Diffie-Hellman or encrypting the key using Asymmetric Cystography such RSA, which is much robust, complex and stronger encryption technic.

I do not understand, RSA stronger than what?  Than DH?  It's not true in general. They are equal from Shor's quantum algorithm point of view.

2. Section 1

Another approach is by establishing a reliable third-party who is generating certificates and encryption keys

It's definitely RSA application.  Certificates and third-party authority (PKI) are just a method to authenticate two parties exchange using asymmetric cryptography.

3. To my mind Section 3.2 does not have any relation to the discussed problem of key exchange via optical channel, which is the title of the paper. It just describes technical info about Intel SGX and includes well-known picture from Intel Developer Forum: Figure 5 is a part of slide https://images.anandtech.com/doci/9687/60.jpg without referring an authorship!

4. Section 4

The idea is to modulate the information in a way that cannot be detected by the human eye, and the information can be decoded only by processing the appropriate signals of the video camera.

It's very weak argument! Leading this idea, any radio channel is secure for sure just because humans are not sensitive to radio waves. Why do not to transfer a key via Bluetooth? Human's eye cannot see the transmission!

5. Also, still no estimations of secure channel throughput, computational processing complexity, amount of extra information used to pass secure bits are provided.

An estimation should be not text, but a number of bits, needed to transfer 1 bit of secure key through video stream.

6. The article still lacks very important part of any applied security system: adversary model.

An example of a possible adversary: a person with highly sensitive photo detector can record blinks of LED not only in the vicinity of mobile device, but from the large distance using optical amplifier with a telephoto lens.  How to prevent such attack to the proposed system?

Or maybe a potential adversary does have a capability of photo detector?  But this is a very weak assumption because modern digital photo camera for $300 has the perfect optic and zoom lens to record secure key transfer blinks from several dozen of meters of mobile device.

Author Response

Dear reviewer 1,

Thanks for the comprehensive and detailed comments, which we accept all of it. Following are our detailed answer:

Section 1

There are many known solutions, such as a specific key

exchange protocol proposed by Diffie-Hellman or encrypting

the key using Asymmetric Cystography such RSA, which is

much robust, complex and stronger encryption technic.

I do not understand, RSA stronger than what? Than DH? It's

not true in general. They are equal from Shor's quantum

algorithm point of view.

Answer: correct, changed accordingly.

Section 1

Another approach is by establishing a reliable third-party who

is generating certificates and encryption keys

It's definitely RSA application. Certificates and third-party

authority (PKI) are just a method to authenticate two parties

exchange using asymmetric cryptography.

Answer: Indeed, the main purpose of CA is authentication. However, since it is done using an alternate route it makes it difficult to be tracked.

Anyway, comment removed

To my mind Section 3.2 does not have any relation to the

discussed problem of key exchange via optical channel,

which is the title of the paper. It just describes technical info

about Intel SGX and includes well-known picture from Intel

Developer Forum: Figure 5 is a part of slide

https://images.anandtech.com/doci/9687/60.jpg without

referring an authorship!

Answer: Accepted. Section 3.2 has been removed

Section 4

The idea is to modulate the information in a way that cannot

be detected by the human eye, and the information can be

decoded only by processing the appropriate signals of the

video camera.

It's very weak argument! Leading this idea, any radio channel

is secure for sure just because humans are not sensitive to

radio waves. Why do not to transfer a key via Bluetooth?

Human's eye cannot see the transmission!

Answer: correct, it is a weak argument. We changed accordingly.

Also, still no estimations of secure channel throughput,

computational processing complexity, amount of extra

information used to pass secure bits are provided.

An estimation should be not text, but a number of bits,

needed to transfer 1 bit of secure key through video stream.

Answer: Indeed, the volume of transmitting video is larger in magnitude from a common encrypted bit stream. However, we concentrate just on key transmission, which is a short message and less frequent. This is a drawback but still justified due to the security it adds and so it is still a valid option to consider.  

In this context we dont consider our approach for transmission of large volumes, which indeed must include predicted impact on the transmission time and volume.

We added to the paper these considerations.

The article still lacks very important part of any applied

security system: adversary model.

An example of a possible adversary: a person with highly

sensitive photo detector can record blinks of LED not only in

the vicinity of mobile device, but from the large distance

using optical amplifier with a telephoto lens. How to prevent

such attack to the proposed system?

Or maybe a potential adversary does have a capability of

photo detector? But this is a very weak assumption because

modern digital photo camera for $300 has the perfect optic

and zoom lens to record secure key transfer blinks from

several dozen of meters of mobile device.

Answer: We added this point and proposed practical solutions

Reviewer 2 Report

I thank the authors for addressing my comments and concerns. However, in my opnion they are only partially covered and some major changes are still required.

Mainly, I still think the experimental validation must be improved. Some standard tests, for example NIST tests, in order to evaluate the performace of cryptographic and security mechanisms are required and essential to undertstand the novelty and beneficts of the proposed technology.

This new experimental section should also include a new and extended discussion section, addressing how the proposed approach solves the previously identified problems. 

Without relevant experiments, the manuscript cannot be accepted. 

Author Response

Dear reviewer 2,

Thanks for the comprehensive and detailed comments, which we accept all of it. Following are our detailed answer:

I thank the authors for addressing my comments and

concerns. However, in my opinion they are only partially

covered and some major changes are still required.

Mainly, I still think the experimental validation must be

improved. Some standard tests, for example NIST tests, in

order to evaluate the performance of cryptographic and

security mechanisms are required and essential to

understand the novelty and benefits of the proposed

technology.

This new experimental section should also include a new and

extended discussion section, addressing how the proposed

approach solves the previously identified problems.

Without relevant experiments, the manuscript cannot be

accepted.

Answer: Key management provides the foundation for the secure generation, storage, use, distribution and destruction of keys [2]. In our experiment we used SGX secured environment to generate the key and save it there. The key is moved to the USB-connected blinking device for encoding the key to a blinking LED stream. The key transmission  has been done using optical transmission mechanism. We used a handy blinking LED device attached to the sender computer via a USB connector. According to the device specifications the blinking speed can be up to 1000 blinks per second. However, the collecting smartphone-camera we used, is limited to 30 blinks per second. Therefore, we have tuned the blinking-device to 30 images per second, which seems enough for the purpose of the experiment to prove the feasibility of using basic wireless optical communications for key transmission, eliminating the need for a permanent, long and costly  setup. It is mobile, available everywhere anytime and requires simple connections to operate.  It is secured and safe in the level outlined in [1], see bellow more details.  It is very similar to the described OC system by Xuhua in [2] figures 4.1 and OCSP in figure 4.2.

The experiment key distribution is done according to the NIST recommendation as described in [2] section 8.1.5.2.2.1 and 2. During distribution the key  must be protected throughout the distribution process, either manual or automated. In case of a manual distribution, keys should be encrypted or be distributed using appropriate physical security procedures. The distribution is from and to authorized sources,  the keys are protected by an approved key-wrapping scheme used only for key wrapping or using a public key-transport key owned by the intended recipient. In our experiment we comply with these guidelines by encrypting the encoded key [blinking LED] by an asymmetric cryptography used just for key wrapping.

In case of distribution via a communication channel, only approved key-wrapping or public key-transport schemes shall be used. For symmetric key-wrapping schemes, the distributed key is not disclosed or modified. For asymmetric key-transport schemes, the private key-transport key and the distributed key are not disclosed or modified, and correct association between the private and public key-transport keys is maintained. The keys are protected in accordance with Section 6. It should provide the following assurances: each entity in the distribution process knows the identifier associated with the other entity, the keys are correctly associated with the entities involved in the key-distribution process, and the keys have been received correctly. Our experiment complies with these guidelines as it is already embedded as a standard in current implementations. For extra protection we use VPN for key distribution.     

[1] Elaine Barker, National Institute of Standards and Technology [NIST] Special Publication 800-57 Part 1 Revision 4, Recommendation for Key Management, Available at http://dx.doi.org/10.6028/NIST.SP.800-57pt1r4, January 2016

[2] Xuhua Wang, Security Performance and Protocol Consideration in Optical Communication System with Optical Layer Security Enabled by Optical Coding Techniques - Spectral Phase Coding (SPC), Sections 3-4. 2015

Reviewer 3 Report

The paper presents the possibility of transferring cryptographic keys using optical communications and inter-device hand-held video transmission. In a scenario presented in the paper, smartphones camera capabilities are used.
The authors have revised the paper in accordance with the reviewers' comments.

Author Response

Dear reviewer 3,

Thanks very much for your positive comment.

Round 3

Reviewer 1 Report

No comments.

Reviewer 2 Report

I think the paper may be accepted in its current form